# Acute Effects of Different Overspeed Loads with Motorized Towing System in Young Athletes: A Pilot Study

**DOI:** 10.3390/biology11081223

**Published:** 2022-08-16

**Authors:** Pau Cecília-Gallego, Adrián Odriozola, José Vicente Beltrán-Garrido, Jesús Álvarez-Herms

**Affiliations:** 1Health and Sport Sciences University School (EUSES), Rovira i Virgili University, 43870 Amposta, Spain; 2National Institute of Physical Education (INEFC), University of Barcelona, 08038 Barcelona, Spain; 3Sport Genomics Research Group, Department of Genetics, Physical Anthropology and Animal Physiology, Faculty of Science and Technology, University of the Basque Country (UPV/EHU), 48940 Leioa, Spain; 4KDNA Genomics®, Joxe Mari Korta Research Center, University of the Basque Country UPV/EHU, 20018 Donostia-San Sebastián, Spain; 5Phymo® Lab, Physiology and Molecular laboratory, 40170 Collado Hermoso, Spain; 6Department of Education and Specific Didactics, Faculty of Humanities and Social Sciences, Universitat Jaume I, 12071 Castellón de la Plana, Spain

**Keywords:** assisted sprint, amateur athletes, individualization, ecological approach, body weight

## Abstract

**Simple Summary:**

The towing system is a widely used method to increase running speed since it allows the athletes to run at a speed higher than the speed that they would achieve with their own methods. Currently, we dispose of motorized towing systems that provide overspeed conditions. In this study, we have analyzed the acute effects of using one of these devices with different loads in eight young athletes and thus determine the possible optimal load for their use during training. These effects have been analyzed on different variables of the sprint running technique, and it has been observed that the increase in speed is mainly due to the increase in stride length, flight time and horizontal distance of the vertical projection of the center of gravity to the first contact of the foot with the ground, as well as to reductions in contact time. However, it cannot be determined whether these results are produced by the muscular action of the athlete or by the action of the device. Following up the investigation of the effects of the motorized towing system and the individualization of loads may allow for the development of these training methods, giving them greater scientific evidence.

**Abstract:**

Overspeed is a training method used to improve running speed, although its effects are not supported by consensual scientific evidence. The overspeed stimulus can be boosted by several methods, including motorized towing devices. Our objectives were to analyze the acute effects of three overspeed loads in young athletes and to select optimal loads for training periods. Eight young athletes (16.73 ± 1.69 years) performed one unassisted sprint and three assisted sprints, and kinematic and biomechanical data were compared. Significant increases (*p* < 0.05) in step velocity and step length were found with 2, 4, and 5.25 kg in maximum running speed, flight time and horizontal distance from the first contact to the vertical projection of the center of mass with 4 and 5.25 kg. Significant time decreases were found in 5 m flying sprint and contact time with 4 and 5.25 kg, and no significant changes were observed in step rate. The individually recommended loads would be between 3.47 ± 0.68% and 6.94 ± 1.35% body weight. Even having limitations, we can understand this work and its results as a pilot study to replicate the methodology and the use of new devices to more broadly investigate the effects of overspeed.

## 1. Introduction

Maximum running speed (MRS) is a specific and essential factor to stimulate specific performance in sprint disciplines in athletics [1,2]. Specific training for sprinters involves regular repetition of the motor pattern of the maximum speed running cycle [1]. Respecting the basic principle of biological stimuli specificity, two training methods for the improvement of MRS have been widely used: (1) those that use resisted speed stimuli, such as uphill running or with an overload on the athlete’s running action through weighted vests or belts or with a towed sled [3], and (2) those that use assisted speed or overspeed (OS) through downhill running [4], high-speed treadmills [5] or towing systems (TS) [1,6,7], which provide assistance for the athlete to run at a speed greater than the maximum possible speed developed by themselves [1,8]. OS stimuli increase the intensity in running movement specificity and physiologically stimulate the neuromuscular system during OS running actions [8,9]. Mainly, it has been hypothesized that OS training creates a “light up” in the central nervous system that can increase the number of neurons recruited and alter the timing of nervous stimuli, which could lead to an improvement in intramuscular coordination, and theoretically, these changes could cause decreases in contact time and increases in stride rate [9,10], but the latter differs in several studies [10,11]. Mero and Komi [12] also state that OS training can improve nerve conduction velocity, which can lead muscle performance to higher levels of performance.

Studies on the subject have described how the application of acute OS with TS stimuli produce alterations in kinematic, biomechanical and physiological factors [1,6,10,12,13]. The potential effects described above reveal that the response would be athlete-dependent and individual, depending on one’s own response capacity and the repetitiveness of the stimuli [1,7,13,14]. In addition, the acute adaptive response capacity would be key to being able to mechanize actions at a higher speed, and it would be efficient in the motor pattern stimulated with OS. In the selection of loads for OS training with TS, the possibility of maintaining each athlete’s race-specific motor pattern and avoiding reactive or harmful processes due to incorrect or inefficient mechanics must always be taken into account [1,7,13]. This could be especially important in young athletes if they have an unstable motor pattern of MRS and is the main reason why the changes that OS conditions produce in biomechanical and kinematic parameters must be controlled [9]. In addition, OS should be applied to mature and experienced athletes in sprint training [9].

The main criterion recommended in the literature for the choice of loads for OS training with TS is based on the percentage of acute improvement of the assisted MRS compared to the unassisted MRS with the recommended variations being between +3–8% [13,15] and +5–10% [6,12]. However, the heterogeneity of the means used to generate OS with TS complicates the standardization and replication of the conditions of each study [7,10,14,15]. Currently there are devices such as the DynaSpeed (Ergotest Technology AS, Langesund, Norway) [7,16] and the 1080 Sprint (1080 Motion, Lidingö, Sweden) [1,17] that allow for precise quantification and monitoring of the traction load.

Based on this background, we present a pilot study, carried out with the 1080 Sprint motorized device, hypothesizing that the results in the analyzed variables will vary depending on the different loads used and that some differences will appear between individuals that will influence the selection of loads for using these methods in the medium or long term. Moreover, individual particularities due to age, biological maturation, training experience in OS stimulus or running sprint model pattern may directly influence the results. The aims were to analyze the acute effects of different OS loads on linear sprint kinematic and biomechanical parameters in young athletes and to quantitatively identify the optimal theoretical tensile loads recommended for use in OS training periods as a proposal to replicate in other studies and training.

## 2. Materials and Methods

### 2.1. Experimental Approach to the Problem

Following a within-subject design [18], all participants were exposed to three OS conditions (OS1 (2 kg), OS2 (4 kg) and OS3 (5.25 kg)) and to a no-OS condition (S0). The values of the different kinematic variables recorded in the OS conditions were compared to the S0 condition. The variables analyzed were time in 5 m flying sprint (T5m), maximum running speed in 5 m flying sprint (V5m), step velocity (SV), step length (SL), step rate (SR), contact time (CT), flight time (FT) and horizontal distance from the vertical projection of the center of mass on the ground to the first contact of the front foot (HD).

Prior to the study, all athletes completed 2–3 familiarization sessions with the 1080 Sprint device. The session began with a standardized warm-up consisting of an 8–10-min easy run, muscle activation exercises, athletic skill exercises, and 2–3 submaximal sprints. They then performed one maximal unassisted sprint and three assisted sprints with progressively higher loads (2 kg, 4 kg, 5.25 kg), recovering for 8–10 min between repetitions.

Anthropometric data were collected for a better description of the sample and to assess the maturity status of the athletes. Data on weight, height, sitting height, and leg length were collected, and the procedures to determine biological maturation were according with that proposed by Mirwald et al. [19].

### 2.2. Participants

The convenience sample of 8 young athletes (3 men/5 women), who compete at the national level in their age categories, was recruited from the Terres de l’Ebre Sports Technification Center. The characteristics of the sample can be consulted in Table 1. Two athletes compete only in sprint events, two compete in sprints and long jump, three in sprints and hurdles and one in sprints and hammer throw. The study was carried out following the protocols of the Declaration of Helsinki and was approved by the Ethics Committee of the University of the Basque Country (M10_2021_191). Informed consent was obtained from all subjects and from their legal guardians in the case of minors.

### 2.3. Methodological Aspects 

The tests were carried out on outdoor synthetic athletics track between 11:30 a.m. and 1:30 p.m. with a temperature of 15 °C. The athletes followed a randomized order in their participation, performing all sprints in the same order (2, 4, 5.25 kg), to progressively increase physical and psychological activation and to reduce the risk of injury. The assistance in the sprint was carried out using the 1080 Sprint device, and the total distance covered was 50 m. During the first 20 m, the athletes started standing without assistance, in the following 30 m the assistance was applied, and after 50 m, they stopped receiving it, decelerating until their movement stopped. The device cable was attached to the athlete with a belt and carabiner, both provided by the manufacturer. The 1080 Sprint device is provided with 90 m of cable that is mechanically rolled or unrolled by a servo motor (2000 rpm G5 Series Motor; OMRON Corp. Kyoto, Japan), controlled by the Quantum software v. 3.9.9.5 (1080 Motion, Lidingö, Sweden). The 1080 Sprint device was placed at a height of 80 cm so that the trajectory of the assistance was as horizontal as possible. Loads can be adjusted between 1 and 15 kg, with 0.1 kg increments, in the isotonic-assisted mode, and it can be determined exactly how many meters to cover with or without mechanical assistance. The equivalences of the assisted overloads on the percentage of body weight (BW) of the athletes can be observed in Table 1.

### 2.4. Data Analysis

T5m and V5m variables were obtained with photoelectric cells from Chronojump (www.chronojump.org/product-category/races/, accessed on 21 January 2020) located at a height of 1 m and connected to a laptop (Toshiba Satellite Pro R50-B-10v, Kyoto, Japan) with the Chronojump software (1.9.0 version, www.chronojump.org/software/, accessed on 21 January 2020) and were recorded between meters 40 and 45 of each sprint [20]. To obtain the SL, CT, FT and HD variables, the attempts were recorded with a Casio Exilim F1 camera (http://arch.casio-intl.com/asia-mea/en/dc/ex_f1/, accessed on 21 January 2020) at 300 fps [21] and analyzed with the Kinovea 2D analysis software (stable version 0.8.15, www.kinovea.org/download.html, accessed on 21 January 2020) [22,23]. The camera was placed perpendicular to the 45 m of the race at a distance of 13 m from the race line and at a height of 1.5 m. The Parallax effect was counteracted by placing references between 40 and 50 m in the projection where athletes actually are shown on camera when they cross that distance [24]. The values of these variables were taken for both legs in two consecutive steps and were presented as the average value of both. For a better location of the center of mass and the metatarsal in the support, markers were placed at both points. The SR (number of steps/step time (CT + FT)) and SV (SL / step time) variables were calculated indirectly. The attempts were analyzed twice during three consecutive supports (2 steps), approximately between 42.5 and 47.5 m.

### 2.5. Statistical Procedures

The normal distribution of the data was checked using the Shapiro-Wilk test. Intra-trial reliability was assessed with the coefficient of variation (CV) and the 2-factor mixed interclass correlation coefficient (ICC) with 95% confidence intervals (CIs). CV values were considered acceptable when CV ≤ 10% [25,26]. The ICC was interpreted as follows: ICC < 0.50 = poor, 0.5–0.74 = moderate, 0.75–0.9 = good, and >0.9 = excellent [27]. To compare the kinematic parameters of the S0, OS1, OS2 and OS3 conditions, as well as the changes in velocity between conditions, a one-way repeated measure analysis of variance (ANOVA) was used. The assumption of sphericity was evaluated using the Mauchly test. When sphericity was violated (*p* ≤ 0.05), the Greenhouse–Geisser correction was applied. When a significant main effect was observed, post hoc comparisons were performed with the Bonferroni correction. Effect sizes (ES) were evaluated using partial eta squared (***η***2p), considering the values of 0.06, 0.06–0.14, and >0.14 as small, medium, and large ES, respectively. Mean difference was obtained, and the standardized mean difference Cohen’s dz effect size was calculated using the formula dz = t/√(n) [28]. The ESs were interpreted as follows: <0.2 = trivial; 0.2–0.6 = small; 0.6–1.2 = moderate; 1.2–2.0 = large; >2.0 = very large. [29]. The level of significance was established at 0.05 for all tests. All statistical analyses were performed with the JASP application for Mac (0.11.2 version; JASP Team (2019), University of Amsterdam, The Netherlands) and IBM SPSS Statistics for Mac (v.25, IBM, New York, NY, USA).

## 3. Results

### 3.1. Reliability Tests

Each parameter had an acceptable intra-trial consistency with CV values < 10% and moderate-to-excellent ICC values ranging from 0.74 to 1.00.

### 3.2. Comparison of Kinematic Parameters between Speed Conditions

Main effects and post hoc test results are reported in Table 2. Significant main effects of condition (*p* < 0.05) were reported in all the variables except for SR; the significant effect sizes observed were large (*η*^2^_p_ = 0.41 to 0.78).

#### Changes in the Variables in Each Condition with Respect to S0

The percentages of change of each condition with respect to S0 are shown in Table 3 and Figure 1.

## 4. Discussion

The first objective of this study was to analyze the acute effects on different kinematic and biomechanical variables of running speed in different young athletes (See Figure 1) produced by OS stimuli with motorized TS. Overall, the results showed significant increases (See Table 2) in V5m, FT and HD with 4 and 5.25 kg; in SL and OS with 2, 4 and 5.25 kg; and significant decreases in T5m and CT with 4 and 5.25 kg, while no significant differences in SR were found with any load.

Regarding the second objective of the study, the recommended load should be set between 2 and 4 kg depending on the individual, taking into account that the percentage of change in the MRS, and the differences in the variables analyzed should lead to the least possible distortion of the natural pattern of speed run.

Although there are large methodological differences in the related literature that make it difficult to compare data [10], we can observe that the percentage increases in V5m and SV obtained in our study (See Table 3) are similar to previous studies: (1) + 14.3% MRS (*p* < 0.01) with 100–150 N [30] and + 4.4% MRS (*p* < 0.001) with 25 N [11]; (2) + 7.5% MRS (*p* < 0.001) with 5 kg [6]; + 9.4% MRS (*p* < 0.001) with 7 kg [1]; + 1.9% with 3 kg, + 3.6% with 4 kg and +4.8% with 5 kg in SV (*p* < 0.05) [16] and with all the loads (3, 4, 5 kg) in SV (*p* < 0.05) [7]. These significant increases in V5m (4 and 5.25 kg) and SV (2, 4 and 5.25 kg), observed in our study (See Table 3) fall within the parameters recommended in the literature [6,12,13,15], and can be explained mainly by significant increases in SL (2, 4 and 5.25 kg) and FT (4 and 5.25 kg) and CT decreases (4 and 5.25 kg). However, in the present study and in the specialized literature, it is not clear whether these differences are produced by greater muscular activity in the athletes or by the athletes’ forward traction produced by the system [10].

Although possible changes in SR produced by OS stimuli could be expected due to coordination and neuromuscular adjustments also dependent on individuality [2,6,9], these have not been demonstrated in previous studies [8,12]. The results previously stated in a study published by our group [10] report data similar to those obtained in the present study (See Table 2), with non-significant reductions with 2 and 4 kg and a slight non-significant increase with 5.25 kg. In this regard, several authors hypothesized about the possible causes: (1) Mero and Komi [12] indicated the possibility that a higher sporting level predisposes to higher SR values, and (2) Van den Tillaar [7] stated that a greater experience with the device could also influence the differences.

The significant increases in the biomechanical variable HD are related to a potential increase in braking forces and a lower specificity of the exercise [15]. In the study by Clark et al. [15] with different loads generated by elastic bands, only the load of 4.7% of the BW led to significant increases in HD (+ 5.7% (*p* < 0.046)); therefore, it is recommended not to use these loads or higher ones. In our study (See Table 2), the 2 kg load led to non-significant increases, unlike the 4 and 5.25 kg loads. Therefore, the recommended loads should be close to 2 kg (3.47% ± 0.68 of BW).

The device used in this study (1080 Sprint) allows for selecting the assistance loads in kg. However, the expression in absolute values in kg does not provide enough information to be able to replicate training situations. For this reason, a way to express the values in a relative manner would be to express them with respect to the % of the BW described here.

In Table 1, we can see the % of BW that each one of the loads implies, divided by gender and for the entire sample. It is observed that the same load implies a higher % of BW for females, as the article by Van den Tillaar [7] also shows, which provides us the data organized by loads and gender (3 kg: 4.0% men to 4.9% women; 4 kg: 5.4% men to 6.6% women; 5 kg: 6.7% men to 8.2% women), and highlights that the use of the same load with different BW and gender should be modified. The values of other studies with motorized TS are 14.8–22.2% [30]; 3.5% [11]; 7.1% ± 0.6% [6]; 9.9% ± 0.9% [1] with a single load and male athletes in all of them, from 4% for 3 kg, 5.3% for 4 kg and 6.7% for 5 kg [16] also only with men.

From the analysis of the acute effects, we should be able to select the minimum load that allows for overcoming the speed barrier without deteriorating the individual running speed pattern [1,7,13], analyzing the differences in SL, SR, CT, FT and HD, which should not differ significantly between assisted and unassisted conditions. Van den Tillaar [7], in line with this statement, also pays special attention to the need for individualization, recommending loads of 4 and 5 kg based on his data. With the results of our study (See Table 2 and Figure 1), the chosen load would be between 2 kg (3.47% ± 0.68% BW) and 4 kg (6.94 ± 1.35% BW), being individually adjusted to each athlete. A weight load of 5.25 kg (9.07 ± 1.52% BW) would be considered excessive.

These recommendations become even more important for the application of OS training in young athletes, mainly due to the risk of injury and to the possibility of not having a stable specific running pattern [9]. Our sample consists of post-PHV and experienced in speed training athletes, as can be seen in Table 1. Only two studies [1,15], under our acknowledgment, have reported results in adolescents. Clark et al. [1] used the 1080 Sprint with a single load of 7 kg (9.9% BW) on 14 males (18.0 ± 2.5 years) of superior athletic level; thus, their results are not fully comparable with ours. The same authors recommend conducting studies with the same device with different ranges of loads, as in our pilot study. In contrast, the study by Clark et al. [15] would allow us to compare the results and conclusions, especially in females, but it is a study carried out with elastic bands. Their recommendations to use 3.8% BW loads, in a sample of six males (19.7 ± 3.7 years) and four females (17.5 ± 3.1 years), are similar to ours.

In addition to the above, some other important methodological aspects in the search for a method appeared: the need to standardize the distances covered, the time of data collection, and the distances covered with and without assistance. Van den Tillaar’s studies [7,16] provide data from a complete race over 60 m, including the acceleration and maximum speed phases, while other studies [1,6,11,30] and ours correspond to the maximum flying speed after previous accelerations of between 20 and 60 m and with data collection areas between 5 and 20 m. Specifically, in our study, the athletes received no assistance during the first 20 m. According to Van den Tillaar [7], the influence of the assisted load is hardly noticeable in the first phase of acceleration, probably due to the posture of the athletes and the acceleration phase. As a limitation to the present study, we observed the lack of a control group that would allow for analyzation of the differences if receiving assistance throughout the race.

Moreover, other limitations of the present study are the heterogeneity in gender, athletic disciplines and sports level. With this pilot study, we propose that other researchers can replicate or modify our methodology to expand knowledge about OS training and the use of these devices. At the same time, the possibility of reproducing the present study in the future with the same sample and athletes will report changes with training experience and biological maturation related to hormonal and strength-specific characteristics.

Although the analysis of the effects of acute exposure to OS does not provide conclusive data on the effectiveness of the method in the medium or long term, it helps us to understand the type of stimulus that TS produces and the loads associated with it on the kinematic parameters of running at maximum speed, highlighting the need for individualization according to gender, age, sports level or experience with this training method.

## 5. Conclusions

The MRS is acutely increased by the OS conditions generated with the 1080 Sprint device, with variations according to the assisted load used. This increase in MRS is mainly attributed to the increase in SL and the reduction in CT, possibly due more to the action of the device than to greater muscular activity of the athlete.

The magnitude of assisted speed loads should be quantified in a standardized and relative way to the individual characteristics of the athlete. One option would be to express them as a % of the BW. The loads should lead to increases in MRS beyond the speed barrier of the athletes, altering their technical running pattern as little as possible.

The use of the 1080 Sprint device and other similar devices is an excellent option to create replicable and adjustable OS conditions, since it allows the loads to be selected in a precise manner and provides immediate results. Due to this, an important qualitative change could be brought in the analysis of OS as an effective training method.

Future investigations should focus on analyzing the individual effects of OS training periods with TS on the improvement of MRS, and on discovering the mechanisms responsible for the possible changes as well as the optimal minimum dose, based on the analysis of electromyographic activity and the application of reaction forces and/or physiological or epigenetic variables.

## Figures and Tables

**Figure 1 biology-11-01223-f001:**
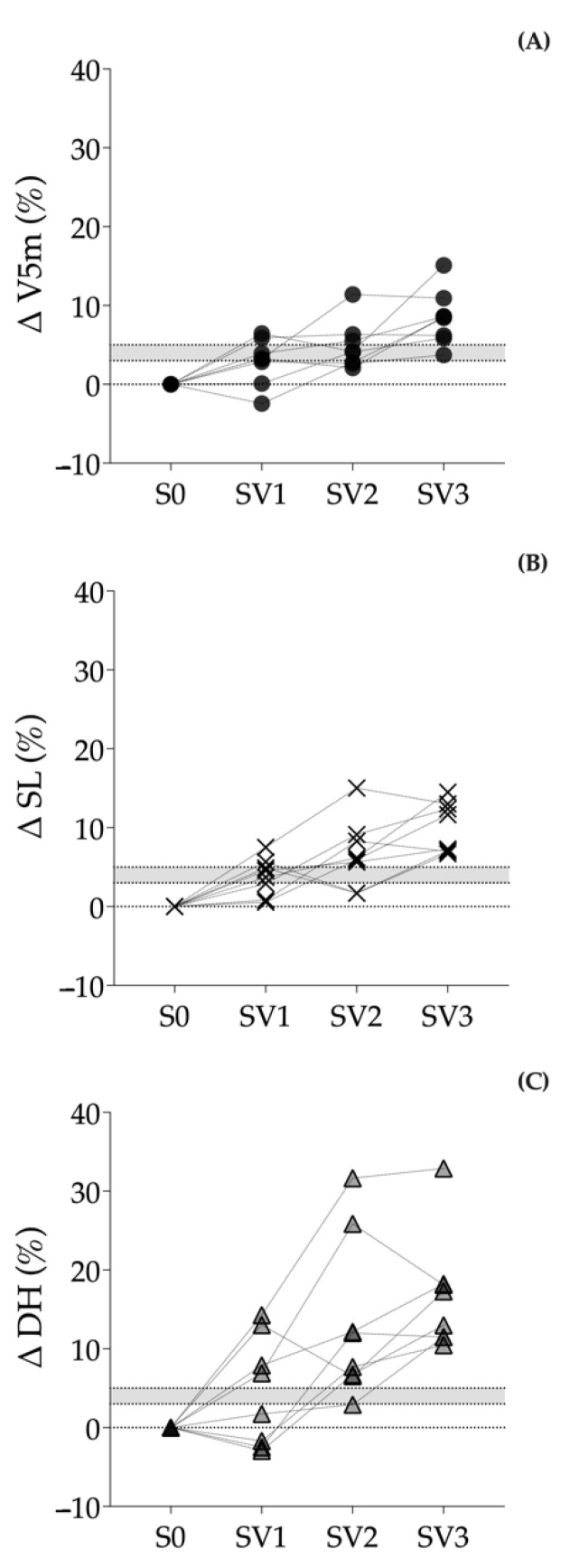
Individual percentages of change for the variables (**A**) V5m, (**B**) SL and (**C**) HD in the different experimental conditions. The gray shaded area highlights the percentage change range between 3% and 5%. V5m: speed between meters 40 and 45 with a flying start; SL: step length; HD: horizontal distance between the point of first contact with the ground and the vertical projection of the center of mass.

**Table 1 biology-11-01223-t001:** Characteristics of the participants and % of body weight in assistance loads.

Sex	Males (3)	Females (5)	All (8)
Age (years)	15.98 ± 1.09	17.18 ± 1.57	16.73 ± 1.69
Years from PHV	2.13 ± 0.68	3.96 ± 1.39	3.28 ± 1.46
Years of training	5.33 ± 1.15	5.60 ± 1.34	5.50 ± 1.20
Height (cm)	174.87 ± 5.78	160.72 ± 13.81	166.03 ± 10.56
Weight (kg)	63.22 ± 2.86	56.62 ± 10.53	59.09 ± 8.70
Fat mass %	7.50 ± 0.17	16.18 ± 6.54	12.93 ± 5.62
% BW OS1	3.17 ± 0.14	3.65 ± 0.84	3.47 ± 0.68
% BW OS2	6.34 ± 0.29	7.31 ± 1.68	6.94 ± 1.35
% BW OS3	8.71 ± 0.40	9.28 ± 1.81	9.07 ± 1.52
PB 60 m (s)	7.63 ± 0.06	8.32 ± 0.60	8.06 ± 0.47

Data are shown as average ± SD. PHV: peak height velocity; BW: body weight; OS: overspeed load (1: 2 kg; 2: 4 kg; 3: 5.25 kg); PB: personal best in 60 m dash with automatic timing.

**Table 2 biology-11-01223-t002:** Kinematic parameters of the different speed conditions.

Variable	S0	O1	OS2	OS3	ANOVA 4 × 1 (*p*)	*η* ^2^ _p_
T5m (s)	0.61 ± 0.04	0.59 ± 0.04	0.58 ± 0.03 * ^L^	0.56 ± 0.05 * ^VL^ # ^L^	<0.001	0.72 ^L^
V5m (m/s)	8.23 ± 0.58	8.46 ± 0.57	8.62 ± 0.48 * ^L^	8.92 ± 0.70 * ^VL^ # ^L^ ^ ^M^	<0.001	0.71 ^L^
SV (m/s)	8.59 ± 0.74	8.88 ± 0.65 * ^L^	9.06 ± 0.71 * ^L^	9.43 ± 0.73 * ^VL^ # ^L^ ^ ^L^	<0.001	0.78 ^L^
SR (step/s)	4.31 ± 0.25	4.30 ± 0.26	4.27 ± 0.28	4.31 ± 0.26	0.753	0.05 ^S^
SL (cm)	199.34 ± 12.36	206.97 ± 13.96 * ^M^	212.70 ± 15.44 * ^L^	219.12 ± 13.75* ^VL^ # ^L^	<0.001	0.75 ^L^
FT (s)	0.119 ± 0.014	0.124 ± 0.014	0.127 ± 0.017 * ^L^	0.125 ± 0.012 * ^S^	0.003	0.47 ^L^
CT (s)	0.114 ± 0.010	0.110 ± 0.007	0.109 ± 0.010 * ^M^	0.108 ± 0.009 * ^L^	0.009	0.41 ^L^
HD (cm)	34.16 ± 4.06	35.60 ± 3.67	38.48 ± 3.81 * ^VL^ # ^VL^	39.72 ± 4.16 * ^VL^ # ^VL^	<0.001	0.74 ^L^

Values are presented as average ± SD. S0: speed without assistance load; OS: assistance loads (1: 2 kg; 2: 4 kg; 3: 5.25 kg); T5m: time between meters 40 and 45 with a flying start; V5m: speed between meters 40 and 45 with a flying start; SV: step velocity between meters 42.5 and 47.5 with a flying start; SR: step rate; SL: step length; FT: flight time; CT: contact time; HD: horizontal distance between the point of first contact with the ground and the vertical projection of the center of mass; * *p_Bonferroni_* ≤ 0.05 different from the S0 condition; # *p_Bonferroni_* ≤ 0.05 different from the OS1 condition; ^ *p_Bonferroni_* ≤ 0.05 different from the OS2 condition; ^T^: trivial effect size; ^S^: small effect size; ^M^: medium/moderate effect size; ^L^: large effect size; ^VL^: very large effect size; ^2^_p_: partial squared eta effect size.

**Table 3 biology-11-01223-t003:** Percentages of change of each condition with respect to S0.

Variables	OS 1–S0	OS 2–S0	OS 3–S0
T5m	−4.80 ± 6.29	−6.64 ± 5.41	−9.55 ± 7.36
V5m	+2.91 ± 2.91	+4.88 ± 3.01	+8.42 ± 3.48
SV	+3.46 ± 2.73	+5.62 ± 3.47	+9.84 ± 3.30
SR	−0.32 ± 2.39	−0.94 ± 3.08	−0.09 ± 1.76
SL	+3.83 ± 2.36	+6.70 ± 4.30	+9.95 ± 3.24
FT	+4.12 ± 2.30	+6.12 ± 4.91	+5.36 ± 3.18
CT	−3.23 ± 3.23	−4.41 ± 3.47	−4.47 ± 3.76
HD	+4.59 ± 6.93	+13.20 ± 10.18	+16.62 ± 7.30

The values are expressed as the % change of each condition with respect to S0. S0: without assistance load; OS: assistance loads (1: 2 kg; 2: 4 kg; 3: 5.25 kg); T5m: time between meters 40 and 45 with a flying start; V5m: speed between meters 40 and 45 with a flying start; SV: step velocity between meters 42.5 and 47.5 with a flying start; SR: step rate; SL: step length; FT: flight time; CT: contact time; HD: horizontal distance between the point of first contact with the ground and the vertical projection of the center of mass.

## Data Availability

In this link, https://drive.google.com/drive/folders/1V1W5-uoHCnKqtMws9OUQqSarTbRbI8BG?usp=sharing, data supporting reported results can be found.

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
