# Peer review of "Acute Effects of Different Overspeed Loads with Motorized Towing System in Young Athletes: A Pilot Study"

_biology, 2022, doi:10.3390/biology11081223_

Round 1
Reviewer 1 Report
The research has an interesting hypothesis. The observed variables fit the hypothesis. The methodological structure is correct even if it is not a "within subject design" model as indicated by the Authors. The number of the sample and sub-groups, divided by gender, is too small. Therefore the results are not generalizable and although statistically supported they don't have the "power" to confirm the hypothesis. Therefore the manuscript is rejected, proposing to resubmit it after an adequate enlargement of the sample involved.
Reviewer 2 Report
I thank the authors for the work done. It is also my opinion that the topic is very interesting.
I will provide suggestions for improving the manuscript step by step. However, it is my opinion that the sample is small, 8 subjects are really few. I would like to suggest to the authors to include the definition "Pilot Study" in the title and in the propose.
Abstract
It is written correctly. Gives highlights from each section of the paper.
Introduction
Line 55: It would be my opinion that, the authors better describe the physiological , neurophysiological and neuromuscular adaptations caused by OS. This would strengthen the whole study in its completeness
Line 75: I would suggest that the authors rephrase the purpose of the study . At the moment it is not very clear, there are multiple objectives, the hypothesis should be moved before the description of the objectives
Materials and Methods
Participants
Line 100: Better describe the characteristics of the enrolled sample (competitors in various track athletics disciplines? ). Experience / years of sports practice? Competitive level of athletes? Athletics disciplines of subjects ?
How did authors decide on the sample size before starting the study? Was there a power analysis?
Discussion
I would include in the limitations of the study the presence of a small sample, also there is no reference to the fact that they are athletes from various track athletics disciplines. Do they have different sports experience?
This could also be an important variable
Reviewer 3 Report
The study seeks to analyze the acute effects of three overspeed loads in young athletes and be able to select optimal loads for training periods. This study analyzed the acute effects of motorized towing systems that provide overspeed conditions with different loads in eight young athletes, and so determine the possible optimal load for their use during training.
The study was conducted on the 1080 Sprint motorized device with two aims: to analyze the acute effects of different OS loads on linear sprint kinematic and biomechanical parameters in young athletes, and to quantitatively identify the optimal theoretical traction loads recommended for use in OS training periods, as a proposed standard model of training.
The practical significance of the study lies in the fact that the use of the 1080 Sprint device and other similar devices is an excellent option to create replicable and adjustable OS conditions, since it allows the loads to be selected in a precise manner and provides immediate results.
As a limitation, the authors noted the lack of a control group that would allow to analyze the differences if receiving assistance throughout the race. In this regard, in the "Discussions" section, it was necessary to reflect the results of other similar experimental studies for comparison.
Overall Recommendation: Accept after minor revision.
Reviewer 4 Report
Dear Authors.
The following is a review of the article entitled "Acute effects of different overspeed loads with motorized towing system in young athletes" which two aims: 1) to analyze the acute effects of different OS loads on linear sprint kinematic and biomechanical parameters in young athletes, and 2) to quantitatively identify the optimal theoretical traction loads recommended for use in OS training periods, as a proposed standard model of training. Thank you very much for thinking of me as a reviewer for this study.
After carefully reading the manuscript, I set forth comments and suggestions for the authors:
Abstract: Correct, but the conclusions could be expanded. Add more practical application and some of the limitations found.
Keywords: Only three key words. Perhaps they could add "amateur athletes"
Introduction: It is necessary to add a paragraph justifying the analysis of the kinematic and biomechanical parameters in young athletes. Reflect if the biological growth and the improvement caused by the maturational development can influence the results of the study (it could be an important limitation).
Materials and Methods: It is recommended to enlarge the sample and to homogenize (only men or only women) or to compare.
The Table 1 “Characteristics of the participants” should show the experience of the athletes.
Expand this information "the competitors in various track athletics disciplines".
The age of the athletes is between 14 and 18 years old. Shouldn't the biological age of the athletes have been controlled? It would be very important to know if the participants are in pre or post pubertal age.
Results: It is not necessary to repeat in text the data we already have in table 2.
Discussion: All figures should be in the results section.
The discussion should focus on comparing the results obtained with the results of other studies in young athletes.
Conclusions: Added practical application.
References: Corrects
Round 2
Reviewer 1 Report
I believe the replies to the reviewer's comments are adequate. In particular, the objective of the study has been downsized, it has been correctly defined as a "pilot study". I would have suggested "preliminary study" for a number of reasons but in fact it has been made clear that the study is "preliminary" and therefore cannot give certain confirmation of the hypothesis. The authors refer to a subsequent study where, I hope, there will be a larger sample and a more rigorous methodology. Therefore it is accepted in the present form.
Reviewer 2 Report
The manuscript has been greatly improved. I would suggest acceptance